# Multiple Sensor Fault Detection Using Index-Based Method

**DOI:** 10.3390/s22207988

**Published:** 2022-10-19

**Authors:** Daijiry Narzary, Kalyana Chakravarthy Veluvolu

**Affiliations:** School of Electronics and Electrical Engineering, Kyungpook National University, Daegu 41566, Korea

**Keywords:** fault detection, fault detection index, residuals analysis, permanent magnet synchronous motor, multi-sensor faults

## Abstract

The research on sensor fault detection has drawn much interest in recent years. Abrupt, incipient, and intermittent sensor faults can cause the complete blackout of the system if left undetected. In this research, we examined the observer-based residual analysis via index-based approaches for fault detection of multiple sensors in a healthy drive. Seven main indices including the moving mean, average, root mean square, energy, variance, first-order derivative, second-order derivative, and auto-correlation-based index were employed and analyzed for sensor fault diagnosis. In addition, an auxiliary index was computed to differentiate a faulty sensor from a non-faulty one. These index-based methods were utilized for further analysis of sensor fault detection operating under a range of various loads, varying speeds, and fault severity levels. The simulation results on a permanent magnet synchronous motor (PMSM) are provided to demonstrate the pros and cons of various index-based methods for various fault detection scenarios.

## 1. Introduction

Sensors are frequently employed to gather data and signals, in particular in the monitoring of electrical devices and drives, the environment, and human health [1,2]. For instance, sensors are used in electrical motor drives to measure and detect changes in position, temperature, displacement, electrical current, as well as many other characteristics [3,4]. However, industrial sensors’ applicability relies on the applications and conditions in which they are utilized. They are required to perform under challenging situations, such as severe and extreme environments with extremely low or high temperatures, vibrations, excessive humidity, etc. [5].

Any industrial drive’s efficiency is entirely dependent on the output of the sensor’s readings. An unexpected variation in the measured signal output, however, may be referred to as a sensor malfunction [6]. There are several causes of sensor faults, including poor manufacturing practices, long-term use wear and tear, and incorrect calibration. This frequently leads to physical divergence from the sensor body’s design parameters, producing misleading and incorrect outputs [7]. Bias, drift, scaling, noise, and hard faults including signal loss are the main causes of sensor malfunction [8]. Different sensors, including voltage, current, temperature, pressure, and position sensors, are typically used in fault detection and diagnosis schemes [9]. The gathered sensor data reveal important details about the system’s health, including whether it is functioning normally or not. Sensor fault diagnosis can be broadly divided into two main categories: hardware method and software method [10]. The hardware method uses multiple components and the same input signals, which are further utilized for comparison, and specific methods such as voting and limit test, etc., are utilized for fault detection. On the other hand, the software method is subdivided into model-based [11], signal-processing-based [12,13], and knowledge-based detection methods [14]. Any type of sensor fault can deteriorate the overall performance of an industrial drive by reducing its reliability. Therefore, it is necessary to investigate the sensor fault diagnosis of the drives, in order to ensure continuous drive operations [15,16]. Model-based methods [17,18,19,20] detect sensor faults by monitoring the residual signal, which is the difference between the real process and the analytical redundancy under normal working conditions. They are considered as the most common techniques in industrial applications. The common residual generation methods include observer-based methods [21,22], parity space methods [23], and parameter identification methods for effectively detecting the sensor faults in satellite control systems and industrial motor drives. However, in signal processing methods for sensor fault detections, a faulty sensor signal of a motor is analyzed with signal processing techniques such as the fast Fourier transform (FFT) [24], wavelet transform (WT) [25], and Hilbert transform [26]. In [27], the STFT and WT methods were used for fault diagnosis such as demagnetization, rotor eccentricity faults, and sensor faults of the servo drive. Recent studies have used the signal processing fault diagnosis techniques focusing on the current, motor vibration, and voltage signals. In [28], the short-time-Fourier-transform (STFT)-based inverter fault detections were used for spectral analysis to detect open-circuit faults in a wind power converter. The knowledge-based method uses the summary of prior knowledge to describe the relationship between the fault and symptom. Interturn short-circuit faults in a drive were detected by using support vector machines (SVMs) and convolutional neural networks (CNNs) in [29]. In [30], bipolar transistor faults, single current sensor fault, and rotor position faults were diagnosed by the FDI algorithm designed by using the SVM technique. In [31], the demagnetization fault was identified using noise and torque information fusion technologies. Similarly, in [32], a Kalman-filter-based sensor fusion method was used to simultaneously measure the three-degree-of-freedom angular displacements and velocity of a ball-joint-like permanent magnet spherical motor.

Nevertheless, so far, the majority of the detection techniques rely only on data from one or more sensors. In the above methods, simultaneous or sequential faults in multiple sensors such as abrupt, incipient, and intermittent faults [33] were not discussed. Hence, in this work, three types of sensor faults such a abrupt, incipient, and intermittent faults are detected by the response of the indices generated from the fusion of speed, current, and voltage sensor residuals. Abrupt faults are modeled as a sudden step-like deviation in which the component value abruptly changes from its nominal value to an unknown faulty one. Incipient faults develop slowly, and intermittent faults usually manifest themselves intermittently in an unpredictable manner. Usually, abrupt faults and incipient faults have a persistent nature, while intermittent faults do not. In this paper, an intermittent fault [34,35] was considered to be periodic with a fixed value. Existing studies employed multiple sensors for the same sensor channel to reduce the noise and improve the fault detection accuracy by sensor fusion. Our proposed approach fundamentally differs in the way that we rely on a single sensor for one sensor channel. As a fault in one sensor channel affects the other sensor channels, we employed the index-based methods to analyze and identify the faulty and healthy channel.

In this work, finite time observers were employed for residual generation for analysis with various index-based methods. The drive was assumed to be healthy, and the issue of faults in multiple sensors was studied in the paper. Multiple sensors’ fault detections based on indices were designed by using the moving root mean square index (MRI), moving-average-based index (MAI), moving-variance-based index (MVI), moving-energy-based index (MEI), first-order-derivative-based index ($DBI_{1}$), second-order-derivative-based index ($DBI_{2}$), and auto-correlation-based index ($A_{c}I$). An auxiliary index (AI) was also developed to select the accurate index values for faulty sensor detections. These index-based techniques provide quick and accurate fault detection. Cost effectiveness was also achieved by the extremely low computational burden of these index-based methods. For evaluation, the index-based fault detection methodology was tested on a permanent magnet synchronous motor (PMSM) with multiple sensors that were employed for speed, current, and voltage measurement. The simulation results are presented together with descriptions of the index-based detections for various defective and noisy settings. A comparative result is also presented to show the efficacy of the proposed method.

## 2. Index-Based Methods

To achieve accurate fault detection in multi-sensor faults, the following indices were considered for the analysis.

### 2.1. Moving-Average-Based Index

The moving average for a signal p(t) can be calculated as follows:(1)$$MAI_{i}=\frac{1}{T_{s}}\sum_{n=i-T_{s}}^{i}p_{i}\left(t\right)$$
where $MAI_{i}$ is the mean of the signal in the *i*th window. $T_{s}$ is the number of samples in one cycle, and *t* is the time step of one sample. Each second moving average for each sample of a signal for the mean of a window of 1 s was calculated on the sample. The transients of the sensor residuals were analyzed by using the index-based methods. The $MAI_{i}$ value remains constant during the PMSM motor’s healthy sensor conditions, but it changes immediately after the fault occurs. Because of this, a threshold was used to compare the $MAI_{i}$ and determine whether the index indicated the presence of faulty sensors. The value of the threshold for this index was considered as 0.5, by using Otsu’s thresholding method.

### 2.2. Moving-RMS-Based Index

For abrupt and incipient sensor faults, the faults on one sensor affect the residuals of the other sensors. Hence, to detect the actual faulty sensor, the moving-RMS-based index is calculated as follows:(2)$$MRI_{i}=\sqrt{\frac{1}{T_{s}}\sum_{n=i-T_{s}}^{i}p_{n}^{2}}$$
where $MRI_{i}$ is the root mean square of the signal p(t) in the *n*th window. In this article, the root mean square was applied to the residuals of the stator currents, speed, and stator voltage values, with the number of samples in one cycle denoted as $T_{s}$ and the time step of one sample as *t*. The object calculates the root mean square (RMS) of the windowed data at each iteration through the window. It can also be seen that the energy of the signal is directly proportional to the MRI values of the residuals, considering a constant window length. Like the behavior of the MAI, the MRI exhibits smooth fluctuations during the healthy sensor state, but it indicates a change during abrupt and incipient faults. The index for MRI, like the index MAI, was compared with a threshold designed using Otsu’s thresholding. The threshold’s considered value was 0.5.

### 2.3. Moving-Variance-Based Index

A variance-based index was used to separate the faulty sensor from the non-faulty ones by comparing it with the set threshold. Utilizing the formula of the moving average, $MVI_{i}$ is considered as follows:(3)$$MVI_{i}=\frac{1}{T_{s}}\sum_{n=i-T_{s}}^{i}({\left(p_{n}-{\bar{p}}_{n}\right)}^{2})$$
where $p_{n}$ denotes each sample of the sensor residual and ${\bar{p}}_{n}$ is the average of the samples of the residuals in the specified window.

The moving variance calculates the variance of the signal around the mean in the given window. When a fault occurs, the abrupt changes cause large deviations, and this affects the variance of the signal.

### 2.4. Moving-Energy-Based Index

Abrupt and incipient faults are also detected by another index, called the moving-energy-based index. The index is calculated and then compared with a threshold.

This index can be calculated as follows:(4)$$MEI_{i}=\frac{1}{T_{s}}\sum_{n=i-T_{s}}^{i}p_{n}^{2}$$
where $MEI_{i}$ is the moving energy of the signal denoted by $S_{n}$. This index, like the MVI, exhibits the same behavior as the previous three indices. The comparison threshold was set at 0.2 and was created using Otsu’s thresholding technique [36]

### 2.5. First-Order-Derivative-Based Index $\left(DBI_{1}\right)$

In this work, residual-based fault analysis was performed by designing a first-order-derivative-based index. It can be calculated as follows:(5)$$DBI_{1i}=\lim_{x_{2}\to x_{1}}\frac{f\left(x_{2}\right)-f\left(x_{1}\right)}{x_{2}-x_{1}}$$

The idea behind using a $\left(DBI_{1}\right)$-based index is to amplify the very slight changes in the faulty sensor values, as well as the noises present in the sensor residual transients. This index was compared to a threshold of 0.5 calculated using Otsu’s thresholding method.

### 2.6. Second-Order-Derivative-Based Index $\left(DBI_{2}\right)$

A second-order-derivative-based index was also designed for the analysis of residuals for fault detections. It can be calculated as shown below:(6)$$DBI_{2i}=\lim_{x_{2}\to x_{1}}\frac{\frac{d}{dx}f(x_{2})-\frac{d}{dx}f(x_{1})}{x_{2}-x_{1}}$$

The $DBI_{2}$ index method, like the $DBI_{1}$ index method, analyzes the transients of noisy abrupt and incipient faults. The threshold was set to 0.5 and was created with Otsu’s thresholding method.

### 2.7. Auto-Correlation Index

Another index utilized here for sensor fault detection was the auto-correlation index. This index was used for analyzing the residuals for intermittent faults. The mathematical expression for $\left(A_{c}I\right)$ is shown as
(7)$$A_{c}I=\frac{\sum_{i=k+1}^{n}\left(y_{i}-\bar{y}\right)\left(y_{i-k}-\bar{y}\right)}{\sum_{i=1}^{n}{\left(y_{i}-\bar{y}\right)}^{2}}$$
where $A_{c}I$ is the auto-correlation of the signal for time series of the signal $y_{i}$, and it lies between −1 and +1. $\bar{y}$ is the overall mean; n is the total number of samples; $y_{i}$ is the value of the signal at sample *i*.

### 2.8. Auxiliary Index

In order to detect the sensor faults more accurately and preserve the reliability of the index-based methods, an auxiliary condition was considered by using an auxiliary index (AI). The mathematical representation of the auxiliary index is as follows:(8)$$AI=V_{indices}\left(f\right)>V_{indices}\left(nf\right)$$
where $V_{indices}$ are MRI, MAI, MVI, MEI, $DBI_{1}$, $DBI_{2}$, and $A_{c}I$, respectively; *f* indicates the faulty and nf indicates the non-faulty value of the indices.

Moreover, for the quantitative analysis of the proposed indices, the two following criteria, the accuracy (*Acc*) and dependability (*Dep*) of the indices, were calculated by using the formulae as follows:(9)$$Acc\%=\frac{\mathrm{Number}\;\mathrm{of}\;\mathrm{correctly}\;\mathrm{detected}\;\mathrm{cases}}{\mathrm{Total}\;\mathrm{number}\;\mathrm{of}\;\mathrm{cases}}$$
(10)$$Dep\%=\frac{\mathrm{Total}\;\mathrm{number}\;\mathrm{of}\;\mathrm{detected}\;\mathrm{faults}\;\mathrm{by}\;\mathrm{the}\;\mathrm{indices}}{\mathrm{Total}\;\mathrm{number}\;\mathrm{of}\;\mathrm{faults}}$$

## 3. Multi-Sensor Fault Diagnosis

In this paper, for the evaluation, we employed the proposed methodology for multi-sensor fault diagnosis in a (PMSM) motor. The dynamics of a PMSM can be modeled as follows: (11)$$\left\{\begin{array}{c} \frac{di_{\alpha }}{dt}=-\frac{R}{L}i_{\alpha }-\frac{1}{L}b_{\alpha }+\frac{1}{L}E_{\alpha }\\ \frac{di_{\beta }}{dt}=-\frac{R}{L}i_{\beta }-\frac{1}{L}b_{\beta }+\frac{1}{L}E_{\beta }\\ \frac{d\omega_{e}}{dt}=-\frac{P}{J}\phi_{e}\left(-\sin\theta_{e}i_{\alpha }+\cos\theta_{e}i_{\beta }\right)-\frac{F_{e}}{J}\omega_{e}-\frac{\Delta_{e}}{J}\\ \frac{d\theta_{e}}{dt}=\omega_{e} \end{array}\right.$$
where $i_{\alpha }$ and $i_{\beta }$ are the stator currents, $E_{\alpha }$ and $E_{\beta }$ are the stator voltages, and $b_{\alpha }$ and $b_{\beta }$ are the back EMFs given as $b_{\alpha }=-K_{e}\omega_{e}\sin\theta_{e}$ and $b_{\beta }=K_{e}\omega_{e}\cos\theta_{e}$, respectively. In the above equations, *R* is the stator resistance, *L* is the synchronous inductance, *P* is the number of pole pairs, *J* is the moment of inertia, $K_{e}$ is the back EMF constant, $\phi_{e}$ is the rotor flux, $F_{e}$ is the viscous friction, $\Delta_{e}$ is the load torque, and $\theta_{e}$ and $\omega_{e}$ are the position and speed of the motor, respectively. The specifications of the motor parameters are defined in Table 1, and the functional block diagram of a PMSM is shown in Figure 1. In this work, abrupt, incipient, and intermittent faults were considered in the speed, current, and voltage sensors of a PMSM. Higher-order sliding mode (HOSM) observers were designed to generate the residuals of the speed and voltage sensors, respectively. However, a Luenberger observer was designed to generate the residuals of the current sensors. The main objective lied in the multi-sensor fault detection of a PMSM. The second objective was to validate the proposed method accordingly.

### 3.1. Generation of Speed Sensor Residuals

In this section, stator voltages ($E_{\alpha }$, $E_{\beta }$) and currents ($i_{\alpha }$, $i_{\beta }$) are considered as known quantities and speed ($\omega_{e}$) is considered as an unknown quantity.
(12)$$\frac{d{\hat{i}}_{\alpha_{1}}}{dt}=-\frac{R}{L}{\hat{i}}_{\alpha_{1}}+\frac{1}{L}E_{\alpha }+\frac{1}{L}\lambda_{1}$$
(13)$$\frac{d{\hat{i}}_{\beta_{1}}}{dt}=-\frac{R}{L}{\hat{i}}_{\beta_{1}}+\frac{1}{L}E_{\beta }+\frac{1}{L}\lambda_{2}$$
where $\lambda_{1}$ and $\lambda_{2}$ are the higher-order terms of STA and can be written as:(14)$$\left\{\begin{array}{c} \lambda_{1}\left(t\right)=-K_{s_{1}}\zeta_{1}\left(\alpha_{s}\left(t\right)\right)-K_{s_{2}}\int_{0}^{t}\zeta_{2}\left(\alpha_{s}\left(t\right)\right)d\tau \\ \lambda_{2}\left(t\right)=-K_{s_{1}}\zeta_{1}\left(\beta_{s}\left(t\right)\right)-K_{s_{2}}\int_{0}^{t}\zeta_{2}\left(\beta_{s}\left(t\right)\right)d\tau \end{array}\right.$$
where $\alpha_{s}$ and $\beta_{s}$ are the selected sliding surfaces and
(15)$$\zeta_{1}\left(\alpha_{s}\left(t\right)\right)=\alpha_{s}\left(t\right)+K_{s_{3}}{\left\lceil \alpha_{s}\right\rfloor }^{\frac{1}{2}}$$
(16)$$\zeta_{2}\left(\alpha_{s}\left(t\right)\right)=\alpha_{s}\left(t\right)+\frac{K_{s_{4}}^{2}}{2}sign\left(\alpha_{s}\left(t\right)\right)+1.5{\left\lceil \alpha_{s}\right\rfloor }^{\frac{1}{2}}$$
where $K_{s_{1}},K_{s_{2}},K_{s_{3}}$, and $K_{s_{4}}$ are properly designed constant terms. Similarly, the terms $\zeta_{1}\left(\beta_{s}\left(t\right)\right)$ and $\zeta_{2}\left(\beta_{s}\left(t\right)\right)$ can be designed by replacing $\alpha_{s}$ with $\beta_{s}$. The estimation error dynamics can be defined as $\alpha_{s}\left(t\right)$ = ${\hat{i}}_{\alpha }-i_{\alpha }$ and $\beta_{s}\left(t\right)$ = ${\hat{i}}_{\beta }-i_{\beta }$ and can be computed similar to [37].

Using the estimated back EMF voltages, the speed of the PMSM can be computed as follows:(17)$${\hat{\omega }}_{e}=\frac{1}{K_{s}}\sqrt{{\hat{b}}_{\alpha }^{2}+{\hat{b}}_{\beta }^{2}}$$
where ${\hat{b}}_{\alpha }=K_{s_{2}}\int_{0}^{t}\zeta_{2}\left(\beta_{s}\left(t\right)\right)d\tau$ and ${\hat{b}}_{\beta }=K_{s_{2}}\int_{0}^{t}\zeta_{2}\left(\beta_{s}\left(t\right)\right)d\tau$, respectively. The speed sensor residuals can be computed as $\omega_{res}$ = ${\hat{\omega }}_{e}-\omega_{e}$.

### 3.2. Generation of Voltage Sensor Residuals

In this section, the stator currents ($i_{\alpha }$, $i_{\beta }$) and speed ($\omega_{e}$) are treated as known quantities and voltages ($E_{\alpha }$, $E_{\beta }$) are treated as unknown quantities. By using the STA-based HOSM observers, the stationary voltages are estimated in the α and β axes, respectively.
(18)$$\left\{\begin{array}{c} \frac{d{\hat{i}}_{\alpha_{2}}}{dt}=-\frac{R}{L}{\hat{i}}_{\alpha_{2}}-\frac{1}{L}b_{\alpha }+\frac{1}{L}\lambda_{3}\left(t\right)\\ \frac{d{\hat{i}}_{\beta_{2}}}{dt}=-\frac{R}{L}{\hat{i}}_{\beta_{2}}-\frac{1}{L}b_{\beta }+\frac{1}{L}\lambda_{4}\left(t\right) \end{array}\right.$$
where $\lambda_{3}\left(t\right)$ and $\lambda_{4}\left(t\right)$ are the gains of the STA observer and can be defined as follows:(19)$$\left\{\begin{array}{c} \lambda_{3}\left(t\right)=-K_{v_{1}}\zeta_{3}\left(V_{1s}\left(t\right)\right)-K_{v_{2}}\int_{0}^{t}sign\left(V_{1s}\left(t\right)\right)d\tau \\ \lambda_{4}\left(t\right)=-K_{v_{1}}\zeta_{4}\left(V_{2s}\left(t\right)\right)-K_{v_{2}}\int_{0}^{t}sign\left(V_{2s}\left(t\right)\right)d\tau \end{array}\right.$$
with
(20)$$\zeta_{3}(V_{1s}\left(t\right))=V_{1s}\left(t\right)+K_{v_{3}}{\left\lceil V_{1s}\right\rceil }^{\frac{1}{2}}$$
(21)$$\zeta_{4}(V_{1s}\left(t\right))=V_{1s}\left(t\right)+\frac{K_{v_{4}}^{2}}{2}sign\left(V_{1s}\left(t\right)\right)+1.5{\left\lceil V_{1s}\right\rceil }^{\frac{1}{2}}$$
where $V_{1s}$ and $V_{2s}$ are the sliding surfaces, respectively, and $K_{v_{1}}$ and $K_{v_{2}}$ are the STA gains. The estimation error dynamics from Equations (Equation 11) and (Equation 18) can be computed from [37].

The unknown voltages can be estimated as follows:(22)$$\left\{\begin{array}{c} {\hat{E}}_{\alpha }=-K_{v_{2}}\int_{0}^{t}\zeta_{2}(V_{1s}\left(t\right))d\tau \\ {\hat{E}}_{\beta }=-K_{v_{2}}\int_{0}^{t}\zeta_{2}(V_{2s}\left(t\right))d\tau \end{array}\right.$$

Hence, the voltage sensor residuals can be computed as $E_{\alpha \beta },res$ = ${\hat{E}}_{\alpha \beta }-E_{\alpha \beta }$.

### 3.3. Generation of Stator Current Sensor Residuals

In this section, the stator currents $\left(i_{\alpha },i_{\beta }\right)$ are treated as unknown quantities and speed $\left(\omega_{e}\right)$ and voltages $\left(E_{\alpha },E_{\beta }\right)$ are treated as known quantities. By using the Luenberger observer, the unknown stator currents are estimated in both the α and β axes. We utilize the PMSM model in the stationary reference frame as
(23)$$\left\{\begin{array}{c} \frac{dx}{dt}=A_{1}x\left(t\right)+B_{1}u\left(t\right)\\ y\left(t\right)=C_{1}x\left(t\right) \end{array}\right.$$
where *x* = [$i_{\alpha },i_{\beta },\omega_{e},\theta_{s}{]}^{T}$ is the state vector. $u={\left[E_{\alpha },E_{\beta },T_{l}\right]}^{T}$ and $y=[\omega_{s},\theta_{s}]$ are the voltages and the input vector, respectively.
$$A_{1}=\left[\begin{array}{cccc} -\frac{R}{L} & 0 & \frac{1}{L}P_{k}sin\theta_{s} & 0\\ 0 & \frac{R}{L} & -\frac{1}{L}P_{k}cos\theta_{s} & 0\\ -\frac{P}{J}\phi_{m}cos\theta_{s} & -\frac{F}{J} & 0 & 0\\ 0 & 0 & 1 & 0 \end{array}\right]$$
$$B_{1}=\left[\begin{array}{ccc} -\frac{1}{L} & 0 & \frac{1}{L}\\ 0 & \frac{1}{L} & 0\\ 0 & 0 & -\frac{1}{J}\\ 0 & 0 & 0 \end{array}\right]$$
$$C_{1}=\left[\begin{array}{cccc} 0 & 0 & 1 & 0\\ 0 & 0 & 0 & 1 \end{array}\right]$$

Th Luenberger observer can be designed as follows:(24)$$\frac{d\hat{x}}{dt}=A_{1}\hat{x}+B_{1}u+L_{t}\left(y-C\hat{x}\right)$$
where $\hat{x}={\left[{\hat{i}}_{\alpha },{\hat{i}}_{\beta },{\hat{\omega }}_{e},{\hat{\theta }}_{s}\right]}^{T}$ is the state estimation vector and $L_{t}$ is the observer gain matrix. The current sensor residuals can be computed as $i_{\alpha \beta ,res}$ = ${\hat{i}}_{\alpha \beta }-i_{\alpha \beta }$.

## 4. Results and Performance Evaluation

This section presents the simulation results to evaluate and demonstrate the effectiveness of the proposed index-based multi-sensor fault detections under different conditions. The specification of the parameters of the PMSM is mentioned in Table 1. The gain values of the HOSM observers for residual generations were selected as shown in [38]. The Luenberger observer (LO) gain matrix, $L_{t}$, can be selected from [39]. As shown in Figure 1, A represents the output of the finite time observers. The generated residuals as shown in Figure 1, B were further used for fault analysis. Hence, the indices MAI, MRI, MVI, MEI, $DBI_{1}$, $DBI_{2}$, and $A_{c}I$ were used for multiple sensors’ fault detections. As shown in Figure 1, C an auxiliary index was used to differentiate the faulty sensor indices (f) from the non-faulty sensor indices (nf). The indices mentioned above can individually detect the sensor faults. Moreover, to improve the reliability and accuracy of the proposed method, the AI was used by collectively considering the indices and differentiating it based on the higher number of either faulty or non-faulty indices. The sensor faults in the PMSM motor can be classified as abrupt faults, incipient faults, and intermittent faults. In order to accurately detect the faults in the sensors, index-based analysis was performed by considering different types of sensor faults.

### 4.1. No-Fault Scenario

As shown in Figure 2, a speed reference of 1000 rpm was considered. The original and the estimated signals of the $i_{\alpha }$ and $i_{\beta }$ currents, speed, and $E_{\alpha }$, $E_{\beta }$ voltage sensors are shown in Figure 2a(i), Figure 2a(ii), Figure 2a(iii), Figure 2a(iv), and Figure 2a(v), respectively. The corresponding residuals for the speed, stator voltages, and stator currents are shown in Figure 2b(i), Figure 2b(ii), Figure 2b(iii), Figure 2b(iv), and Figure 2b(v), respectively.

### 4.2. Multi-Sensor Fault Scenario

In this section, multi-sensor faults are considered in the speed, current, and voltage sensors of a PMSM motor. In the first case, low-severity α, β abrupt current sensor faults with 15% load and low speed were introduced at *t* = 0.739 s and *t* = 1.52 s, as shown in Figure 3a(i) and Figure 3a(ii), respectively. The responses of the speed and voltage sensors are shown in Figure 3a(iii), Figure 3a(iv) and Figure 3a(v), respectively. The residuals of the α-, β-axis current, speed, and α-, β-axis voltage are shown in Figure 3b(i), Figure 3b(ii), Figure 3b(iii), Figure 3b(iv), and Figure 3b(v), respectively. It can be seen that the residuals cross the respective thresholds, indicating a faulty sensor. The residuals of the speed and α-axis voltage lie below the threshold. However, the residuals of the β-axis voltage cross the threshold and indicate a fault due to α- and β-axis current faults. Hence, the residuals were further analyzed using various index-based methods to detect the faulty sensors. The index-based methods the moving root mean square index (MRI), moving-average-based index (MAI), moving-variance-based index (MVI), moving-energy-based index (MEI), first-order-derivative-based index ($DBI_{1}$), second-order-derivative-based index ($DBI_{2}$), and auto-correlation-based index ($A_{c}I$) were designed to detect the faulty sensors in a multi-sensor fault scenario.

As shown in Figure 4a(i), the MRI of the $i_{\alpha }$ crosses its threshold and indicates that $i_{\alpha }$ is faulty at t=0.739. However, the $i_{\alpha }$ residual lies below the threshold, as shown in Figure 4a(ii). The MAI failed to detect the low-severity current faults in the PMSM. The MVI, MEI, $DBI_{1}$, and $DBI_{2}$, however, characterized the faulty condition, as shown in Figure 4a(iii), Figure 4a(iv), Figure 4a(v), and Figure 4a(vi), respectively, at t=0.740 s, 0.7402 s, 0.391 s, and 0.7391 s. Using Equation (Equation 8), it is clear that n=5 and n=1, clearly indicating that the number of fault detection indices was greater than the indices that failed to detect the faults. From Figure 4, it can be seen that, due to the fault in the α-axis of the stator current, the energies of the $I_{\alpha_{res}}$ and the MRI values were proportionally related, for a constant number of samples in the specified moving window. Hence, it can be seen that after the fault, the residual value increased tremendously, leading to an increase in the energy and, hence, and increase the MRI values also. Similarly, the residuals of $i_{\beta }$ were analyzed by using the index-based methods, as shown in Figure 4b. The depicted MRI for the $i_{\beta }$ crosses the threshold at t=1.525 s and indicates a faulty $i_{\beta }$ sensor, as shown in Figure 4b(i). However, the depicted MAI lies below the threshold, as shown in Figure 4b(ii). The MVI, MEI, $DBI_{1}$, and $DBI_{2}$ of the $i_{\beta }$ residuals lie above the threshold and indicate a faulty $i_{\beta }$ sensor, as shown in Figure 4b(iii), Figure 4b(iv), Figure 4b(v), Figure 4b(vi), respectively, at at t=1.528 s, 1.526 s, 1.522 s, and 1.522 s. The AI of $i_{\beta }$ also shows that f=5 and nf=1, hence indicating a faulty $I_{\beta }$ sensor. Similarly, the indices for the $\omega_{e}$ residual are plotted in Figure 5a. The depicted MRI of the ω touches the threshold slightly, as shown in Figure 5a(i). The indices MAI, MVI, MEI, $DBI_{1}$, and $DBI_{2}$ are shown in Figure 5a(ii), Figure 5a(iii), Figure 5a(iv), and Figure 5a(v), respectively. The residuals of $E_{\alpha }$ were also analyzed using the index-based methods of MRI, MAI, MVI, MEI, $DBI_{1}$, and $DBI_{2}$, respectively as shown in Figure 5b(i), Figure 5b(ii), Figure 5b(iii), Figure 5b(iv), Figure 5b(v), and Figure 5b(vi). The MRI, MAI, MVI, MEI, $DBI_{1}$, and $DBI_{2}$ for the residuals of the $E_{\beta }$ voltage sensors lie below the threshold, as shown in Figure 6. From the AI analysis, it can be depicted that $E_{\beta }$ is fault free.

In the second case, a combination of an incipient and an abrupt fault was also introduced in $i_{\alpha }$ and $E_{\beta }$ at *t* = 1.20 s and 1.60 s, respectively, as shown in Figure 7a(i) and Figure 7a(v). A load change of 50% was also considered while introducing the $i_{\alpha }$ and $E_{\beta }$ faults. Due to the faults in both sensors, $i_{\beta }$, $E_{\alpha }$, and $E_{\beta }$ also were affected, as shown in Figure 7a(ii), Figure 7a(iii), and Figure 7a(iv), respectively. The residuals of the $i_{\alpha }$ current sensor cross the threshold at *t* = 1.20 s, as shown in Figure 7b(i). Similarly, the residuals of $i_{\beta }$, $\omega_{e}$, and $E_{\alpha }$ are shown in Figure 7b(ii), Figure 7b(iii), and Figure 7b(iv), respectively. The $E_{\beta }$ residual crosses the threshold and indicates a faulty $E_{\beta }$ sensor, as shown in Figure 7b(v). A further analysis was performed to detect the actual faulty sensor by using the index-based analysis methods.

An interturn fault in the $i_{\alpha }$ sensor can lead to a high fault current in the shorted circuit, which can produce excessive heat and ripples in the torque. The MRI of the $i_{\alpha }$ sensor indicates a fault at t=1.258 s, as shown in Figure 8a(i). However, the MAI lies below the threshold, and hence, it was unable to detect the $i_{\alpha }$ fault, as shown in Figure 8a(ii). The depicted MVI touches the threshold, indicating a fault at *t* = 1.257 s, as shown in Figure 8a(iii). The MEI, $DBI_{1}$, and $DBI_{2}$ of the $i_{\alpha }$ residual increase and cross the threshold at 1.257 s, 1.2563 s, and 1.2563 s, as shown in Figure 8a(iv), Figure 8a(v), and Figure 8a(vi), respectively. The AI was calculated to check the faulty sensor, and it can be seen that n=5 and nf=1; hence, $i_{\alpha }$ was considered as a faulty sensor. The index-based analysis for the $i_{\beta }$ sensor is shown in Figure 8b. The indices (MRI, MAI, MVI, MEI, $DBI_{1}$, and $DBI_{2}$ lie below the selected threshold, as shown in Figure 8b(i), Figure 8b(ii), Figure 8b(iii), Figure 8b(iv), Figure 8b(v), and Figure 8b(vi)), respectively.

The speed ($\omega_{s}$) residual was also analyzed by using the index-based methods, as shown in Figure 9a. The MRI, MAI, MVI, MEI, $DBI_{1}$, and $DBI_{2}$ lie below the threshold, as shown in Figure 9a(i), Figure 9a(ii), Figure 9a(iii), Figure 9a(iv), Figure 9a(v), and Figure 9a(vi), respectively. The analysis of the $E_{\alpha }$ residual was also performed using the index-based methods. The depicted MRI, MAI, MVI, MEI, $DBI_{1}$, and $DBI_{2}$ lie below the selected threshold, which indicates that $E_{\beta }$ is non-faulty, as shown in Figure 9b(i), Figure 9b(ii), Figure 9b(iii), Figure 9b(iv), Figure 9b(v), Figure 9b(vi), respectively.

However, the calculated MRI for the $E_{\beta }$ residual surpasses the threshold at *t* = 1.60 s, indicating a faulty sensor, as shown in Figure 10i. The MAI and MVI, however, stay below the threshold and discriminate the change as a healthy $E_{\beta }$ sensor, as shown in Figure 10ii and Figure 10iii, respectively. However, as shown in Figure 10iv, the MEI for the $E_{\beta }$ residual slightly exceeds the threshold at t=1.62 s and indicates that $E_{\beta }$ is faulty. The $DBI_{1}$ and $DBI_{2}$ of the $E_{\beta }$ sensor residual also increase and cross the threshold at 1.605 s and 1.605 s, as shown in Figure 10v and Figure 10vi, respectively. The AI was thus calculated to further analyze the index-based methods. It can be seen that n=4 and nf=2; hence, $E_{\beta }$ was considered as a faulty sensor.

In the third case, the effect of the speed sensor fault in addition to the current sensor fault was analyzed for accurate fault detections. In this regard, an incipient fault was introduced in $i_{\alpha }$ sensor and an abrupt fault in $\omega_{e}$ sensor at *t* = 1.25 s and *t* = 1.50 s, respectively, as shown in Figure 11. A random noise of 20% was also introduced in the $i_{\alpha }$ sensor. The actual and the estimated states are shown in Figure 11a, and the residuals are shown in Figure 11b, respectively. The MRI, MVI, MEI, $DBI_{1}$, and $DBI_{2}$ lie above the threshold and indicate a faulty sensor at 1.26 s, 1.28 s, 1.48 s, 1.253 s, and 1.253 s, as shown in Figure 12a(i), Figure 12a(iii), Figure 12a(iv), Figure 12a(v), and Figure 12a(vi), respectively. However, the MAI values lie below the threshold, as shown in Figure 12a(ii). Furthermore, it can be seen that f=5 and nf=1. Hence, the $i_{\alpha }$ sensor was concluded to be faulty. Similarly for the $i_{\beta }$ index analysis, all the indices, the MRI, MAI, MVI, MEI, $DBI_{1}$, and $DBI_{2}$, lie below the threshold, as shown in Figure 12b. Hence, $i_{\beta }$ is not faulty. The analysis of ω was also performed in a similar manner.

As shown in Figure 13a, it can be seen that the MRI, MAI, MVI, MEI, $DBI_{1}$, and $DBI_{2}$ cross the threshold, and hence, the abrupt ω fault can be detected, as shown in Figure 13a(i–vi). However, in the case of index analysis method application for the $E_{\alpha }$ sensor, the MRI showed a slow increase in its values and slightly touches the threshold, as shown in Figure 13b(i). The other indices still remain below the threshold. Further utilizing the AI analysis, it can be seen f=1 and nf=5, which implies that f<nf; hence $E_{\beta }$ is non-faulty. Similarly, the index method for $E_{\beta }$ sensor analysis is shown in Figure 14. The calculated AI shows that f=1 and nf=5, and hence, f<nf indicates that $E_{\beta }$ is fault free.

To show the efficacy of the proposed method, an incipient fault in $i_{\alpha }$ sensor was considered at t=1.25 s, and and intermittent fault was considered in the $\omega_{e}$ sensor, with the first fault occurring at t=0.5 s and ending at t=0.8 s. The second intermittent fault occurred at t=1.5 s and ended at t=1.8 s, as shown in Figure 15a. The illustration of the HOSM observer for the $i_{\alpha }$ and $\omega_{e}$ faults is shown in Figure 15b. The $A_{c}I$-based index was calculated to analyze the residuals of the faulty sensors. Using this $A_{c}I$-based method, it can be seen that the incipient fault in the $i_{\alpha }$ sensor failed to be detected, as shown in Figure 16i, whereas the intermittent faults in the $\omega_{e}$ sensor were detected with a delay, as shown in Figure 16iii.

## 5. Discussion

In this research, we took into account both sudden and incipient sensor faults in industrial drives in order to perform multi-sensor fault detection. Different conditions were considered by sequentially introducing the faults in each sensor. Indices (MRI, MAI, MVI, MEI, $DBI_{1}$, and $DBI_{2}$) were designed to detect the faults by selecting a particular threshold. The threshold for the indices was designed by using Otsu-based thresholding techniques.

Different cases such as low-severity current faults, sudden speed changes, and changes in the load were considered on the basis of index analysis for detecting the single- and multi-sensor faults. As shown in Figure 4a, during low-severity $i_{\alpha }$ and $i_{\beta }$ faults, the MRI, MVI, and MEI detected the fault after a certain delay. However, the indices $DBI_{1}$ and $DBI_{2}$ detected the faulty $i_{\alpha }$ sensor with a minimum delay compared to the other indices. Similarly, in Figure 4b, the MRI, MVI, and MEI experienced a certain delay in detecting the $i_{\beta }$ fault. On the other hand, $DBI_{1}$ and $DBI_{2}$ detected the faulty $i_{\beta }$ sensor immediately after the fault’s occurrence. The indices for $\omega_{e}$, $E_{\alpha }$, and $E_{\beta }$ did not show any sudden change and indicated fault-free sensor data, by holding the property of the auxiliary index.

In Figure 8a, due to the load change in the drive, the MRI, MVI, MEI, $DBI_{1}$, and $DBI_{2}$ detected the fault of the $i_{\alpha }$ sensor data. The indices $DBI_{1}$ and $DBI_{2}$, however, detected the fault with a minimum delay compared to the other indices. The $E_{\beta }$ sensor fault in Figure 8 also shows a variation and crosses the respective thresholds when the MRI, MEI, $DBI_{1}$, and $DBI_{2}$ were utilized. The auxiliary index (AI) plays a vital role in selecting the accurate fault without discriminating the outputs of the indices.

A random noise of 20% was also introduced in the $i_{\alpha }$ sensor, as shown in Figure 11a(i). Due to noise in the sensors, the MRI value detected the $i_{\alpha }$ sensor fault at t=1.26 s. The MVI also detected the fault at t=1.30 s with the maximum delay. The $DBI_{1}$ and $DBI_{2}$ detected the noisy $i_{\alpha }$ sensor at t=1.26 s and t=1.262 s, respectively. As shown in Figure 13, all seven indices, MRI, MAI, MVI, MEI, $DBI_{1}$, $DBI_{2}$, and $A_{c}I$, detected the abrupt speed ($\omega_{e}$) fault. As shown in Figure 17, the accuracies of the indices were calculated, and it can be seen that the $A_{c}I$ had the lowest accuracy for various fault conditions.

Similarly, the dependability of the indices was calculated for the comparison of all the indices under various conditions, as shown in Figure 18.

The indices were calculated for all five sensors under three different cases mainly, low-severity current faults, sensor faults during sudden load changes, and impacts of sensor speed changes along with other sensors. A random noise of more than 20% was also tested in the sensors for multiple fault detections. However, due to increase in the noise, the MRI increased and crossed the threshold, indicating a false faulty sensor. The proposed index method can be improved by using a low-pass filter for signals with noises greater than 20%, as the $DBI_{1}$ and $DBI_{2}$ became more sensitive to noise, compromising the fast detection property compared with other indices. Hence, in this case, adaptive thresholds can also be incorporated to prevent the system from false faulty sensor signal detection.

## 6. Conclusions

In this work, different types of sensor faults were analyzed for fault detection by using multiple index-based methods for a healthy drive. Seven index-based methods were analyzed for the identification of the changes that occurred in the faulty sensors and the non-faulty sensors. The results showed that the MRI, MEI, MVI, $DBI_{1}$, and $DBI_{2}$ were able to detect the low-severity faults. The combination of both incipient current and an abrupt voltage fault during the load change could be detected by the MRI, MEI, $DBI_{1}$, and $DBI_{2}$ accurately. The seven index-based methods could also be used to identify variations in the speed sensor when they were combined with a fault in the current sensor. A combination of an intermittent fault in the speed sensor and an incipient fault in the beta-axis of the current sensor was also simulated, and the index-based methods were able to identify the faulty sensors. The simulated results conducted on various fault scenarios showed that index-based analysis can be employed for fast fault detection. Future work will focus on experimental validation of the proposed method on a PMSM motor.

## Figures and Tables

**Figure 1 sensors-22-07988-f001:**
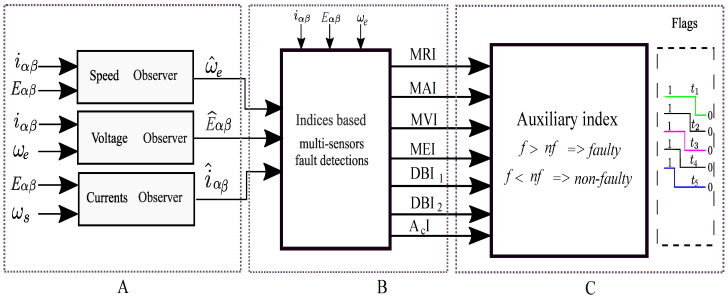
Functional block diagram for multi- sensor fault detection in a PMSM; (**A**): Finite time observer block; (**B**): Indices based detections using the residuals; (**C**): Fault Identification with the AI.

**Figure 2 sensors-22-07988-f002:**
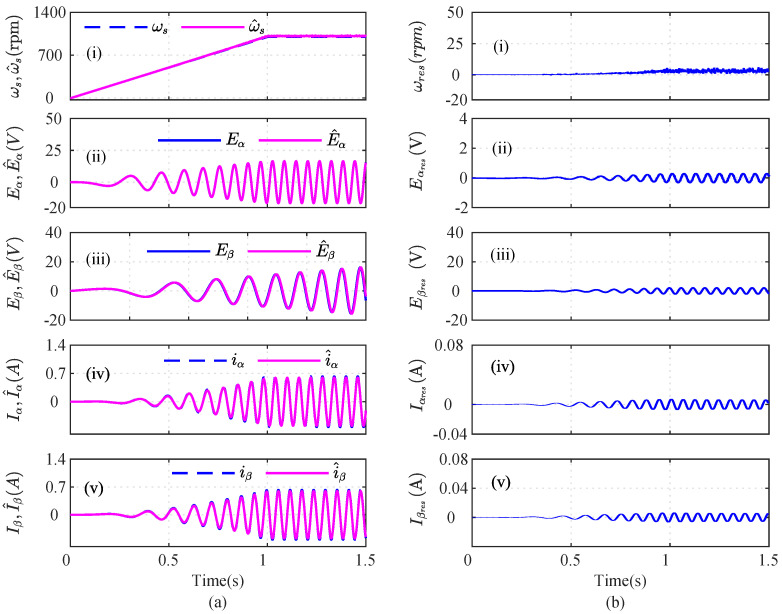
Illustration of current, speed, and voltage sensors’ observers during no-fault scenario; (**a**) actual and estimated signals; (**b**) residuals.

**Figure 3 sensors-22-07988-f003:**
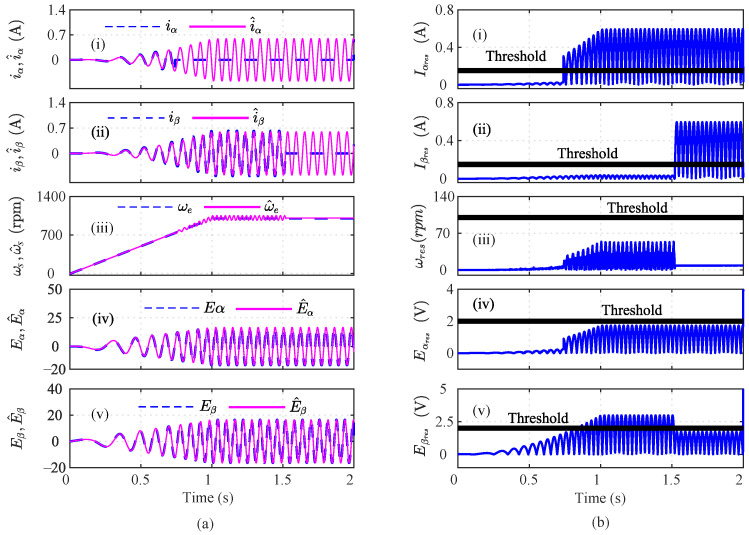
Illustration of current, speed, and voltage sensors’ observers during an abrupt $i_{\alpha }$ and $I_{\beta }$ fault scenario at *t* = 0.739 s and *t* = 1.52 s; (**a**) actual and estimated signals; (**b**) residuals.

**Figure 4 sensors-22-07988-f004:**
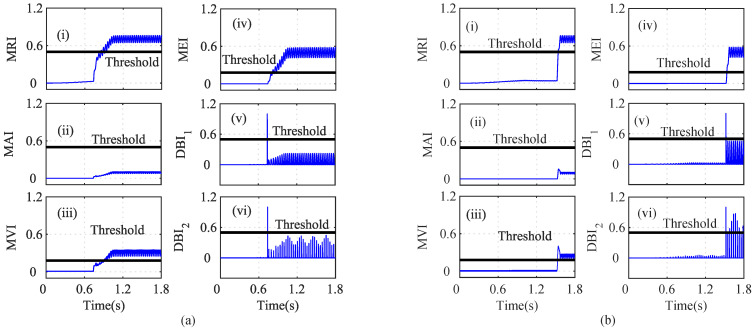
Analysis of residuals for (**a**) $I_{\alpha }$ and (**b**) $I_{\beta }$ using index-based methods ((i) MRI, (ii) MAI, (iii) MVI, (iv) MEI, (v) $DBI_{1}$ and (vi) $DBI_{2}$) for the $I_{\alpha }$ and $I_{\beta }$ faults at *t* = 0.739 s and *t* = 1.52.

**Figure 5 sensors-22-07988-f005:**
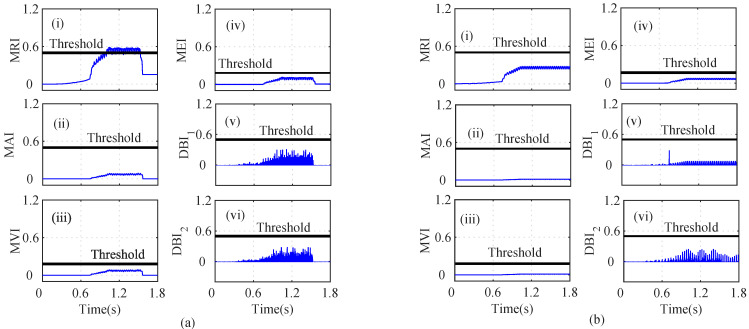
Analysis of residuals for (**a**) ω and (**b**) $V_{\alpha }$ using index-based method ((i) MRI, (ii) MAI, (iii) MVI, (iv) MEI, (v) $DBI_{1}$ and (vi) $DBI_{2}$) for the $I_{\alpha }$ and $I_{\beta }$ faults at *t* = 0.739 s and *t* = 1.52 s.

**Figure 6 sensors-22-07988-f006:**
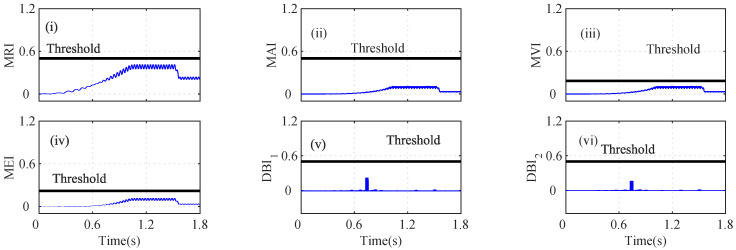
Analysis of residuals for $V_{\beta }$ using index-based method ((i) MRI, (ii) MAI, (iii) MVI, (iv) MEI, (v) $DBI_{1}$ and (vi) $DBI_{2}$) for the $I_{\alpha }$ and $I_{\beta }$ fault at *t* = 0.739 s and *t* = 1.52 s.

**Figure 7 sensors-22-07988-f007:**
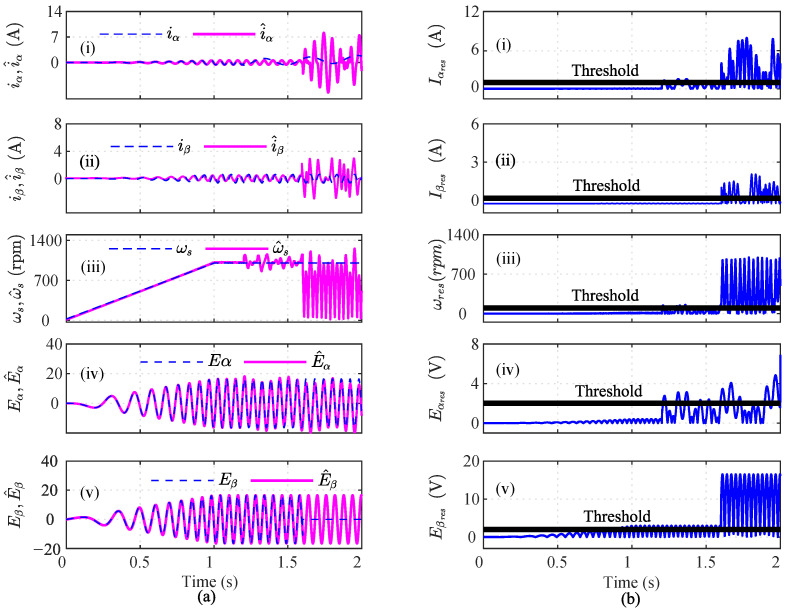
Illustration of current, speed, and voltage sensors’ observers during the incipient $i_{\alpha }$ fault and abrupt $V_{\beta }$ fault scenario at *t* = 1.256 s and *t* = 1.60 s, respectively; (**a**) actual and estimated signals; (**b**) residuals.

**Figure 8 sensors-22-07988-f008:**
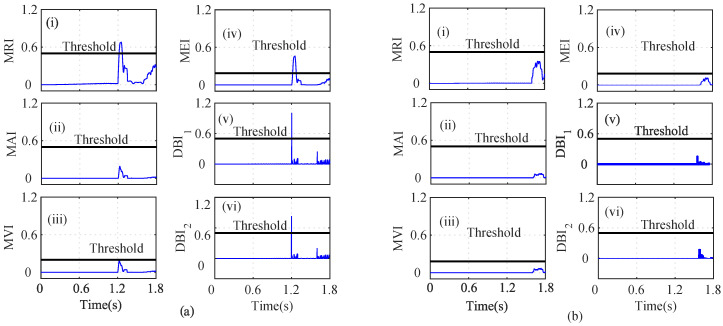
Analysis of residuals for (**a**) $I_{\alpha }$ and (**b**) $I_{\beta }$ using index-based methods ((i) MRI, (ii) MAI, (iii) MVI, (iv) MEI, (v) $DBI_{1}$ and (vi) $DBI_{2}$) during the incipient $i_{\alpha }$ fault and abrupt $V_{\beta }$ fault scenario at *t* = 1.256 s and *t* = 1.60 s, respectively.

**Figure 9 sensors-22-07988-f009:**
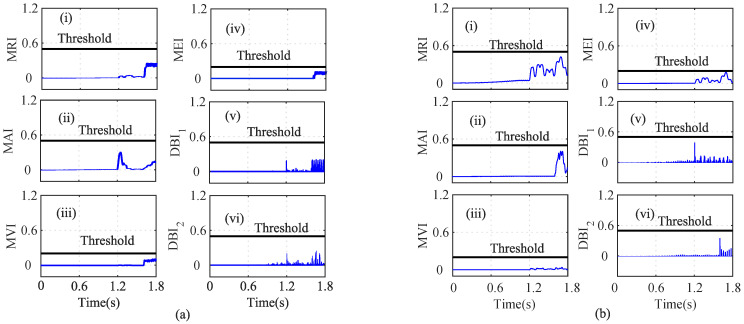
Analysis of residuals for (**a**) *W* and (**b**) $V_{\alpha }$ using index-based method ((i) MRI, (ii) MAI, (iii) MVI, (iv) MEI, (v) $DBI_{1}$ and (vi) $DBI_{2}$) for incipient $I_{\alpha }$ fault and abrupt $V_{\beta }$ fault at *t* = 1.256 s and *t* = 1.60 s.

**Figure 10 sensors-22-07988-f010:**
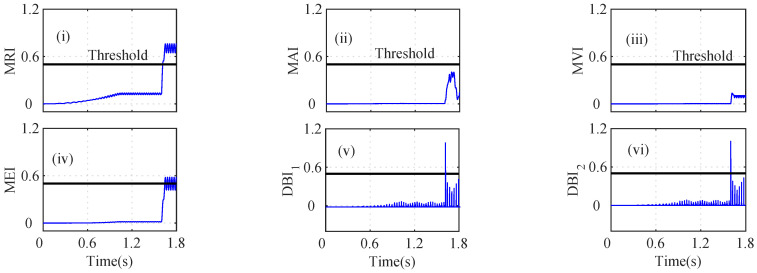
Analysis of residuals for $V_{\beta }$ using index-based method ((i) MRI, (ii) MAI, (iii) MVI, (iv) MEI, (v) $DBI_{1}$ and (vi) $DBI_{2}$) for incipient $I_{\alpha }$ fault and abrupt $V_{\beta }$ fault at *t* = 1.256 s and *t* = 1.60 s.

**Figure 11 sensors-22-07988-f011:**
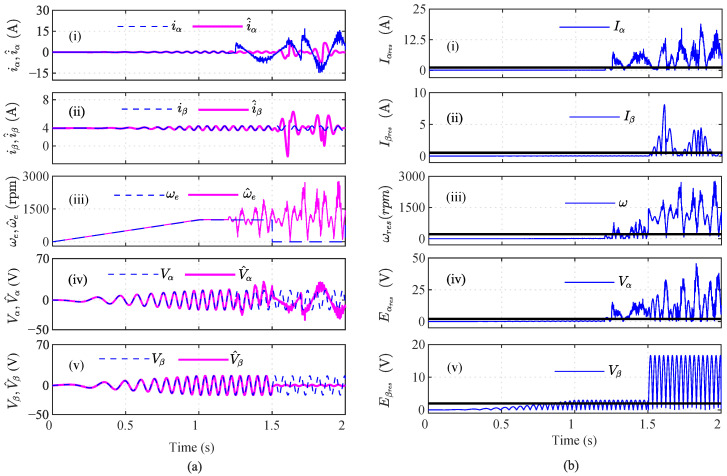
Illustration of current, speed, and voltage sensors’ observers during the 20% noisy incipient $i_{\alpha }$ fault and abrupt $W_{e}$ fault scenario at *t* = 1.25 s and *t* = 1.50 s, respectively; (**a**) actual and estimated signals; (**b**) residuals.

**Figure 12 sensors-22-07988-f012:**
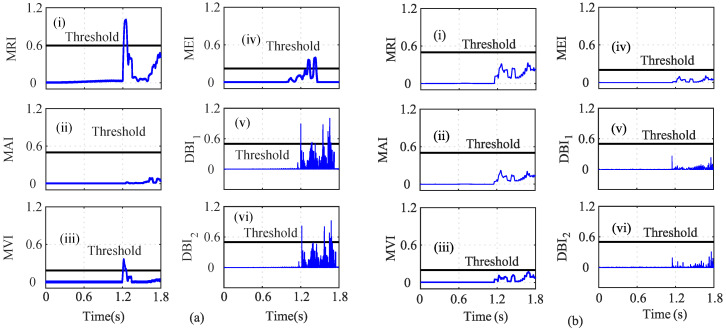
Analysis of residuals for (**a**) $i_{\alpha }$ and (**b**) $i_{\beta }$ using index-based method ((i) MRI, (ii) MAI, (iii) MVI, (iv) MEI, (v) $DBI_{1}$ and (vi) $DBI_{2}$) for the 20% noisy incipient $i_{\alpha }$ fault and abrupt $W_{e}$ fault scenario at *t* = 1.25 s and *t* = 1.50 s, respectively.

**Figure 13 sensors-22-07988-f013:**
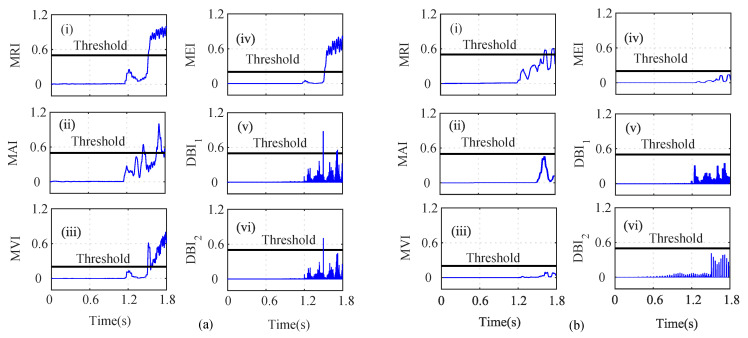
Analysis of residuals for (**a**) $\omega_{e}$ and (**b**) $E_{\alpha }$ using index-based method ((i) MRI, (ii) MAI, (iii) MVI, (iv) MEI, (v) $DBI_{1}$ and (vi) $DBI_{2}$) for the 20% noisy incipient $i_{\alpha }$ fault and abrupt $\omega_{e}$ fault scenario at *t* = 1.25 s and *t* = 1.50 s, respectively.

**Figure 14 sensors-22-07988-f014:**
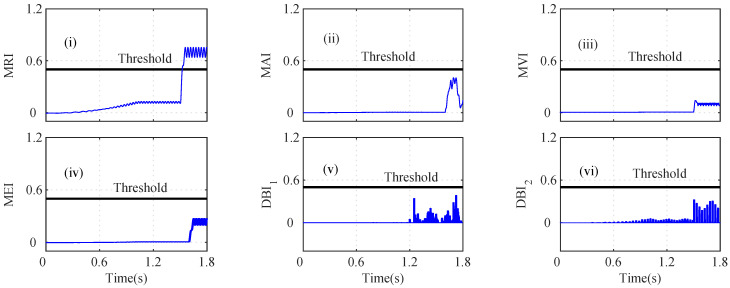
Analysis of residuals for $E_{\beta }$ using index-based method ((i) MRI, (ii) MAI, (iii) MVI, (iv) MEI, (v) $DBI_{1}$ and (vi) $DBI_{2}$) for the 20% noisy incipient $i_{\alpha }$ fault and abrupt $\omega_{e}$ fault scenario at *t* = 1.25 s and *t* = 1.50 s, respectively.

**Figure 15 sensors-22-07988-f015:**
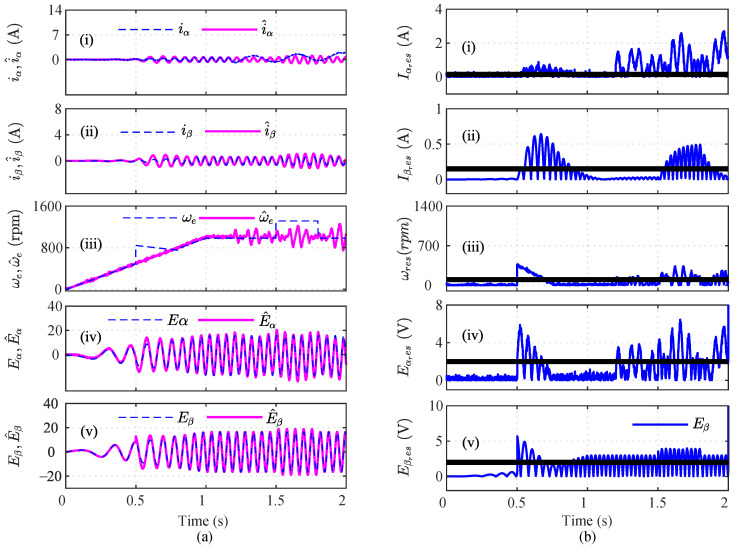
Illustration of current, speed, and voltage sensors’ observers during the incipient $i_{\alpha }$ fault at *t* = 1.25 s and intermittent $W_{\omega }$ fault scenario at *t* = 0.5 s and *t* = 1.50 s, respectively; (**a**) actual and estimated signal; (**b**) generated residuals.

**Figure 16 sensors-22-07988-f016:**
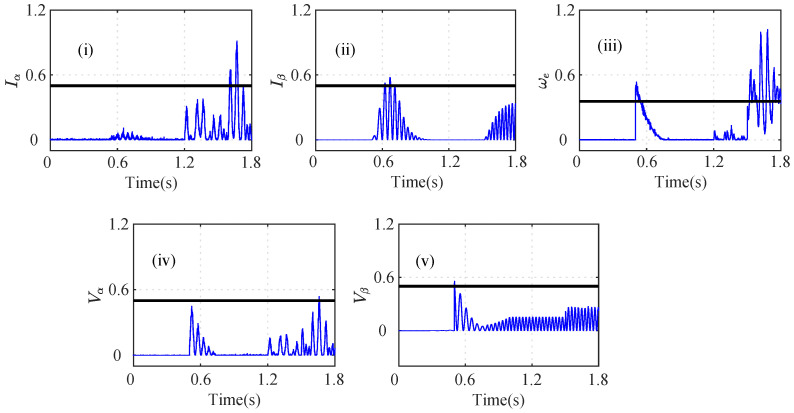
$A_{c}I$-based analysis for current ((i) $I_{\alpha }$, (ii) $I_{\beta }$), speed ((iii) $\omega_{e}$), and voltage sensors’ ( (iv) $V_{\alpha }$, (v) $V_{\beta }$) residuals during the incipient $i_{\alpha }$ fault at *t* = 1.25 s and intermittent $W_{\omega }$ fault scenario at *t* = 0.5 s and *t* = 1.50 s, respectively.

**Figure 17 sensors-22-07988-f017:**
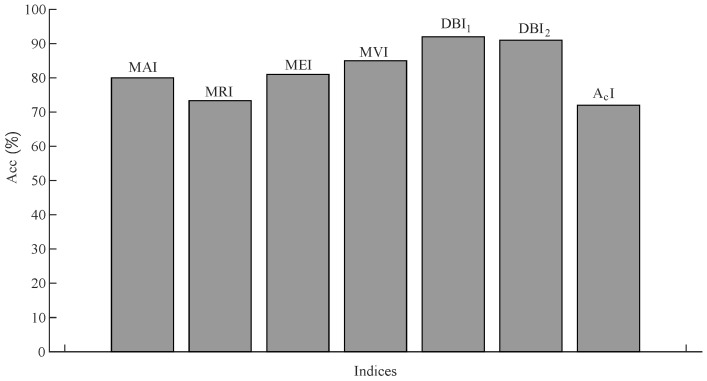
Accuracies of the indices used in the proposed method.

**Figure 18 sensors-22-07988-f018:**
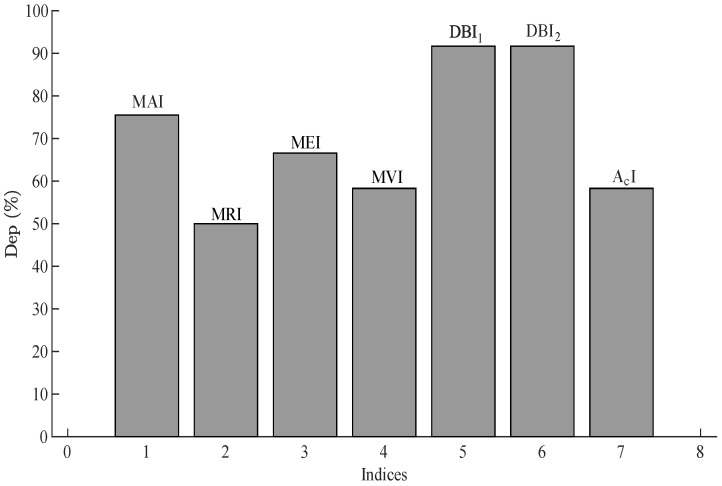
Dependability of the indices used in the proposed method.

**Table 1 sensors-22-07988-t001:** Specifications of the PMSM.

Quantity	Symbol	Value
PMSM Rating	$P_{w}$	50 (kW)
Rating speed	$\omega_{e}$	628 (rad/s)
Stator inductance	*L*	0.47 (mH)
Stator resistance	*R*	0.79 (Ω)
Magnetic flux linkage	$K_{e}$	0.2709 (Vs/rad)
Number of poles	*P*	4

## Data Availability

Not applicable.

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
