# Peer review of "Multiple Sensor Fault Detection Using Index-Based Method"

_sensors, 2022, doi:10.3390/s22207988_

Round 1

Reviewer 1 Report

This is a very important topic and a well-written paper. Here are my comments:

·        Intermittent failures are sometimes defined as those occurring from time to time but their frequency increases till they become a permanent failure. Is this the type of failure you are referring to?

·        You should clarify in the Introduction the difference between a failure in the drive that is detected by an error-free sensor and an error-free drive but the sensor observing the drive, fails.

·        Using data from several sensors is often called sensor fusion. You should relate your methodology to sensor fusion.

·        If the beginning of your contribution is in line 156, please disregard this comment. Otherwise, clearly indicate the beginning of your contribution.

·        Line 162: Table 1. Where is it? I am guessing it is in line 179. If I am correct, it should be moved up and clearly identified.

·        Please fix the format in lines 173-175

·        There is problem in line 250: “par”

·        The random noise is 20%. Why specifically 20?

In summary, a very good paper in my opinion but after some minor changes as indicated above.

Author Response

  1. Intermittent failures are sometimes defined as those occurring from time to time but their frequency increases till they become a permanent failure. Is this the type of failure you are referring to?

Reply: The authors thank the reviewer for their comment. Intermittent faults [1], [2] occur in an unexpected, intermittent manner. In rare instances, it is conceivable for the same sort of fault to occur several times at varying intervals. It might also occur many times. In this paper, an intermittent fault is considered to be periodic with a fixed value.   We have now revised the content in the Introduction (Page 2, Lines  71-76 )

References

[1]   M. Yu and D. Wang, "Model-Based Health Monitoring for a Vehicle Steering System With Multiple Faults of Unknown Types," in IEEE Transactions on Industrial Electronics, vol. 61, no. 7, pp. 3574-3586, July 2014.

[2] D. Zhou, Y. Zhao, Z. Wang, X. He and M. Gao, "Review on Diagnosis Techniques for Intermittent Faults in Dynamic Systems," in IEEE Transactions on Industrial Electronics, vol. 67, no. 3, pp. 2337-2347, March 2020.

  1. You should clarify in the Introduction the difference between a failure in the drive that is detected by an error-free sensor and an error-free drive but the sensor observing the drive, fails.

Reply: The authors would like to thank the reviewer for the comment.

In this work, fault detection of multiple sensors faults based on the observer based residual analysis is designed by indices-based methods for a healthy drive.  The faults are assumed to be occurring in the sensors alone and drive is assumed to be error-free and healthy during the fault detection.

We have now revised the content to address the issue and avoid confusion. Abstract (Page 1, Lines 2-4), introduction (Page 2, Lines 85-92) and conclusion (Page 18, Lines 357-360) are revised accordingly.

.

  1. Using data from several sensors is often called sensor fusion. You should relate your methodology to sensor fusion.

Reply:  The authors would like to thank the reviewer for the comment.

Various sensors of a complex system, requires many sensors for accomplishing the role to continuously drive the complex system. And sensor fusion [1], [2] technique to reduce the noise, requires multiple sensors, which can be costly and thus increases the complexity of the sensors. However, the proposed work employs sensor data from other sensors of the system to identify the fault using indices-based methods.

Existing studies  employed multiple sensors for the same sensor channel to reduce the noise and to improve the fault detection accuracy by sensor fusion. Our proposed approach fundamentally differs in the way that we rely on a single sensor for one sensor channel. As fault in one sensor channel effects the other sensor channels, we employ the indices based methods to analyze and identify the faulty and healthy channel.

We have now revised  the content in the introduction section. (Page 02).

References:

[1] M. Zhu, W. Hu and N. C. Kar, "Multi-Sensor Fusion-Based Permanent Magnet Demagnetization Detection in Permanent Magnet Synchronous Machines," in IEEE Transactions on Magnetics, vol. 54, no. 11, pp. 1-6, Nov. 2018.

[2] M. Li, K. -M. Lee and E. Hanson, "Sensor Fusion Based on Embedded Measurements for Real-Time Three-DOF Orientation Motion Estimation of a Weight-Compensated Spherical Motor," in IEEE Transactions on Instrumentation and Measurement, vol. 71, pp. 1-9, 2022.

  1. If the beginning of your contribution is in line 156, please disregard this comment. Otherwise, clearly indicate the beginning of your contribution.

Reply: The authors thank the reviewer for the comment. The contribution of this work is now clearly stated in Page 02, Lines 76-88.

  1. Line 162: Table 1. Where is it? I am guessing it is in line 179. If I am correct, it should be moved up and clearly identified.

Reply:  The authors thank the reviewer for the comment. The table is corrected in the revised paper (Table 01, Page 7)

  1. Please fix the format in lines 173-175.

Reply: Thank you for raising this concern. We have revised the content to address the issue.

  1. There is problem in line 250: “par”

Reply: Thank you for the comment. It is now corrected in the main text.

  1. The random noise is 20%. Why specifically 20?

Reply: Thank you for this comment. Random noises of 5 %, 10 %, 15 % and 20 % are considered in this work. The sensor fault is detected by using Otsu’s Thresholding based Technique for the six indices MRI, MAI, MEI, MVI,  and . Six different thresholds are considered for sensor fault detections by using those indices. A random noise of more than 20 % is also tested in the sensors for multiple fault detections. However, using the indices-based methods with the set thresholds, the proposed method fails to detect the faults above the noise level 20 %. Hence, this work is limited only to sensor faults with noise level 20 %. However, the work will be expanded in the future for fault detections of sensors with higher noise levels.  We have now revised the content to include the above discussion in the paper ( Page 16, lines 347-352)

Reviewer 2 Report

Dear Authors,

I greatly appreciate the work that was done and interesting methodology you applied. I see great potential of the paper, but unfortunately in this form it cannot be considered for publication. In particular,

- English style is poor, terminology inadequate, text is sometimes completely incomprehensible and misleading, starting from the title itself,

- starting from the Introduction, faults of the device are not distinguished from the faults of the sensors that supervise the device, which makes very difficult to understand what it all is about,

- the object, purpose and means must be clearly defined and consistently kept, which was not done,

- it is not clear, what failure is spoken about, that of device or sensor,

- it is not clear, is the study based on simulations only, or any sort of experiments was made (it should be clearly stated in "materials and methods" section and consistently described),

- some less important errors are present, like insufficient explanation of the equation (line 154), extraordinary letters (lines 179, 250), lack of the table caption (line 180), wrong section divisions  (lines 95, 149), etc. These makes it difficult to understand the idea and arguments presented. 

Detailed remarks can be found in the attached file.

Thus, I recommend to reject the paper in this form, but cordially encourage the Authors to revise the paper thoroughly, considering my suggestions and to submit the revised, complete version once again.

Author Response

  1. English style is poor, terminology inadequate, text is sometimes completely incomprehensible and misleading, starting from the title itself.

Reply: The authors thank the reviewer for the comment. Authors have taken utmost care to address all the grammatical and typographical errors in the paper.

  1. starting from the Introduction, faults of the device are not distinguished from the faults of the sensors that supervise the device, which makes very difficult to understand what it all is about,

Reply: The authors would like to thank the reviewer for the comment.

In this work, fault detection of multiple sensors faults based on the observer based residual analysis is designed by indices-based methods for a healthy drive.  The faults are assumed to be occurring in the sensors alone and drive is assumed to be error-free and healthy during the fault detection.

We have now revised the content to address the issue and avoid confusion. Abstract (Page 1, Lines 2-4), introduction (Page 2, Lines 85-92) and conclusion (Page 18, Lines 357-360) are revised accordingly.

  1. the object, purpose and means must be clearly defined and consistently kept, which was not done

Reply: The authors would like to thank the reviewer for the comment. We have now revised the content in the Abstract, Introduction to define the objective and purpose of the paper.

  1. it is not clear, what failure is spoken about, that of device or sensor.

Reply: The authors would like to thank the reviewer for the comment.

In this work, fault detection of multiple sensors faults based on the observer based residual analysis is designed by indices-based methods for a healthy drive.  The faults are assumed to be occurring in the sensors alone and drive is assumed to be error-free and healthy during the fault detection.

We have now revised the content to address the issue and avoid confusion. Abstract (Page 1, Lines 2-4), introduction (Page 2, Lines 85-92) and conclusion (Page 18, Lines 357-360) are revised accordingly.

5. some less important errors are present, like insufficient explanation of the equation (line 154), extraordinary letters (lines 179, 250), lack of the table caption (line 180), wrong section divisions  (lines 95, 149), etc. These makes it difficult to understand the idea and arguments presented. 

Reply: Thank you for the comments. The comments along with the line numbers are addressed accordingly.

 As mentioned in line 154,   is calculated individually for each term (MRI, MVI, MEI,   and  )

The extra letters in line no 179 and 250 are removed in the main revised file.

The table caption is included in the revised version on Page 7, Table 1.

The section divisions in lines 95 and 149 are corrected in the revised file.

Reviewer 3 Report

Dear authors, I have the following comments:

1.      Lines 156 (PMSM) and 185 (HOSM), The first abbreviation should be accompanied by its full name.

2.      Almost all parts of the blue curves in Figure 3 are obscured by the red curves, so it is recommended to adjust their spatial relative position.

3.      Simulation verification alone is not enough. Considering the complexity of reality, the method proposed in this paper should be validated at least at the experimental bench level.

4.      Lack of comparative experiments. The method of this paper should be compared with existing methods.

Author Response

  1. Lines 156 (PMSM) and 185 (HOSM), The first abbreviation should be accompanied by its full name.

Reply: The authors would like to thank the reviewer for this comment. The full form of the abbreviation of PMSM in line 156, is mentioned in line 95  in the main revised file. Similarly, the full form of the abbreviation of HOSM in line 185 is now mentioned in line 173  in the main text file. Both the modified terms are highlighted in color red in the main revised file.

  1. Almost all parts of the blue curves in Figure 3 are obscured by the red curves, so it is recommended to adjust their spatial relative position.

Reply: The authors would like to thank the reviewer for their comment. The magenta color curve is re-drawn with twice the size of the blue curve in all the figures  as shown in Figure  2(Page 8), Figure 3(Page 9), Figure 7(Page 12) and Figure 11(Page 14) respectively in the revised version.

  1. Simulation verification alone is not enough. Considering the complexity of reality, the method proposed in this paper should be validated at least at the experimental bench level.

Reply: The authors thank the reviewer for the comment. This proposed work is limited to simulations of more than one sensor faults introduced separately and simultaneously. However, in the future, the simulation results will be validated in the experimental set up with more conditions imposed to detect various faults.

  1. Lack of comparative experiments. The method of this paper should be compared with existing methods.

Reply: The authors thank the reviewer for the comment. The authors have introduced another indices called autocorrelation-based index () to detect faults in both the current and voltage sensor faults.  The formula for the  auto-correlation based index  and details are included in the revised file in Page 4, lines 155-158.

An incipient fault is introduced in the alpha axis of the stator current at t=1.25 s and an intermittent fault in the speed sensor occurring at two time intervals, with the first one in the duration of t=0.5 s to t=0.8 s; second one in the duration of t=1.5 s and t=1.8 s respectively as shown in Figure 15 on Page 16 in the revised file. The HOSM observer is utilized to generate the residuals for the current, voltage and speed observers. The residuals are then utilized for further analysis of exact faulty sensors. Using this  based method, it can be seen that the incipient fault in the  sensor fails to get detected, whereas the intermittent faults in the  sensor gets detected with a delay in it as shown in Figure 16 on Page 16 in the revised file.

The accuracies of all the indices are also calculated to show the fault detection efficiency of the indices. The accuracy of the  based index is less as compared to the other indices used in the proposed method working under various load and noise conditions as shown in Figure 17 on Page 18 in the revised version.

Round 2

Reviewer 2 Report

Dear Authors,

thank you very much for the thorough revision of the submitted text. I recommend it for publishing after minor revision. Please note at least three main suggestions:

- English style should be improved (e.g. in line 3, "fault" is twice repeated making the thing unclear, "fault detection of multiple sensors faults," and, similarly, in lines 3-4, "based" is repeated three times, "sensors faults based on the observer based residual analysis by indices based methods," there are some unclear and misleading formulations in Conclusions, etc.).

- Putting units in the brackets should be unified for the entire manuscript, since in Tab. 1 they are in square brackets, while in Figures in rounded ones. Similarly, numbering of the graphs in figures divided in (a) and (b) parts, should be consistent.

- 28 references out of 34 are from IEEE journals. Even though it is very renowned database, it seems to be one-sided presentation of the state-of-art, since there are many journals dealing with sensors issued by Springer, T&F, Elsevier and other publishers. Please, broaden your literature review.

Please find some detailed remarks in the attached file. 

Thank you very much for the good job.

Author Response

  1. - English style should be improved (e.g. in line 3, "fault" is twice repeated making the thing unclear, "fault detection of multiple sensors faults," and, similarly, in lines 3-4, "based" is repeated three times, "sensors faults based on the observer based residual analysis by indices based methods," there are some unclear and misleading formulations in Conclusions, etc.).

Reply: The authors would like to thank the reviewer for their comment. We have now revised the content to improve the clarity and avoid confusion. The content in the  Abstract, Conclusion are revised and the rest of the paper are revised accordingly.  and in the Conclusion Section in the main file.

  1. - Putting units in the brackets should be unified for the entire manuscript, since in Tab. 1 they are in square brackets, while in Figures in rounded ones. Similarly, numbering of the graphs in figures divided in (a) and (b) parts, should be consistent.

Reply: The authors thank the reviewer for the comment. We have now revised the content in the main file to avoid inconsistent notations.  The numbering of the graphs is modified and updated in the revised version. Figure 15 is updated and divided in sections (a) and(b) and the context is highlighted in lines 313-320 in the revised version.

  1. - 28 references out of 34 are from IEEE journals. Even though it is very renowned database, it seems to be one-sided presentation of the state-of-art, since there are many journals dealing with sensors issued by Springer, T&F, Elsevier and other publishers. Please, broaden your literature review.

Reply: The authors thank the reviewer for the comment. We have now included more references from other publishers to broaden the literature review.

Reviewer 3 Report

1. The meaning of index AI should be further explained.

2. The procedure of the fault detection should be further described or drawn to clearly indicate how to use the indices to identify a fault.

3. It is more convincing to include at least two or more indicators for comparison.

Author Response

  1. The meaning of index AI should be further explained.

Reply: The authors would like to thank the reviewer for their comment. The auxiliary index (AI) is used to differentiate the actual faults by observing the values of the indices used. We have now included detailed information on the AI index in the revised paper.

The number of faulty indices and the non- faulty indices values are compared and the ones with the higher values are considered as either faulty or non-faulty sensor by using the equation no. 8 as shown in Page no. 4.  This content is revised and highlighted in the main text file as shown in lines 153-158.

  1. The procedure of the fault detection should be further described or drawn to clearly indicate how to use the indices to identify a fault.

Reply: The authors would like to thank the reviewer for their comment. In the revised paper, we have included an overview block diagram for the proposed scheme. 

The overall architecture of fault diagnosis scheme is shown in Figure 1 (Page 6). As shown in Figure 1, section A, represents the output of the finite time observers. The generated residuals in section B, are further used for fault analysis. Hence, indices such as MAI, MRI, MVI, MEI, ,  and  are used for multiple sensor fault detections. A threshold is also designed by using Otsu’s thresholding technique for fault detections. The value of the indices are compared with the respective thresholds and the higher values of indices as compared with the thresholds indicate a faulty sensor.  As shown in section C, an auxiliary index is used to differentiate the faulty sensor indices( f)  from the non-faultu sensor indices (nf).  The indices mentioned above can individually detect the sensor faults. For better analysis, the accuracies and dependability of the proposed indices based methods are calculated.

 The content is now presented in the revised file in lines 193-207.

  1. It is more convincing to include at least two or more indicators for comparison.

Reply: The authors would like to thank the reviewer for their comment.

All the indices/ indicators mentioned in this work can be employed independently to detect faults. However, to improve the reliability of the proposed method, the auxiliary index (AI) is employed to improve the collective performance. In the revised paper, we have now compared the performance of all the indices independently together with the auxiliary index.

To improve the dependability and accuracy of the proposed method, AI is used by collectively considering the indices and differentiating it based on the higher number of either faulty or non-faulty indices. The number of indices with higher values indicates that the sensor is either faulty or non-faulty.

The authors have now revised the content in the main file in lines 343-352.